# Determining the Use and Reasons for Non-De-Escalation of Empiric Carbapenem Therapy in a Private Hospital in South Africa

**DOI:** 10.3390/antibiotics14121220

**Published:** 2025-12-03

**Authors:** Petro de Klerk, Lindi A. Zikalala-Mabope, Phumzile P. Skosana

**Affiliations:** 1Department of Clinical Pharmacy, School of Pharmacy, Sefako Makgatho Health Sciences University, Molotlegi Street, Ga-Rankuwa, Pretoria 0208, South Africa; pbester66@gmail.com; 2Office of the Dean, School of Pharmacy, Sefako Makgatho Health Sciences University, Molotlegi Street, Ga-Rankuwa, Pretoria 0208, South Africa; lindi.zikalala@smu.ac.za

**Keywords:** carbapenem, cultures, de-escalation, empiric therapy, targeted therapy, South Africa

## Abstract

Background: Due to the rising incidence of ESBL-producing bacterial infections, the use of carbapenems has increased over recent decades. Carbapenems are part of the group of last-resort antimicrobials and are used widely as empirical therapy, which is contributing to the growing rate of antimicrobial resistance (AMR). De-escalation has been proven to be a successful tool in antimicrobial stewardship programmes (ASPs) in minimising the occurrence of AMR and decreasing the use of antimicrobials. The purpose of the study was to find the reasons why prescribers do not de-escalate from empiric carbapenem therapy. Methods: This retrospective quantitative study was conducted in a private hospital in South Africa. The infection markers and cultures of these patients were considered. Results: De-escalation was practiced in 17% of the patients. Empiric carbapenem therapy was started in 11.2% of patients and the most prescribed carbapenem was ertapenem (62.4%). Cultures were available in 71.1% of the study population. De-escalation was not performed in 83% of patients, mostly since their infection markers decreased with carbapenem therapy (45.9%) or because of culture unavailability (28.9%). Conclusion: The study came to the conclusion that prescribers do not want to de-escalate once their patients are improving on current treatment or if there are no cultures available.

## 1. Introduction

Antimicrobial resistance is defined by the World Health Organization (WHO) as a conversion that occurs when microorganisms are treated with antimicrobial drugs, which leads to the antimicrobials becoming ineffective [1]. The WHO states that current antibiotics are becoming very sparse due to the growing rates of AMR worldwide and since very few new antibiotics are being produced [1]. According to an analysis that was performed on the burden of AMR globally, it was found that the most prevalent resistant organisms which led to the largest number of deaths were *Escherichia coli*, followed by *Staphylococcus aureus*; *Klebsiella pneumoniae*; *Streptococcus pneumoniae*; *Acinetobacter baumannii*; and *Pseudomonas aeruginosa* [2]. It has been identified, globally, that Enterobacteriaceae has started to develop resistance to carbapenems and to other antimicrobials, e.g., fluoroquinolones, aminoglycosides and sulphonamides, making it harder to treat these infections [3].

In the surveillance report released by the National Department of Health (NDoH) of South Africa in 2024, it was reported that the most common organism cultured from blood in the private and public sectors was *K. pneumoniae*, followed by *S. aureus*, *E. coli*, *A. baumannii* and then *Enterococcus faecalis*. This report also states that in the last five years, there was an average of 70% prevalence of ESBL-producing *K. pneumoniae* organisms in South Africa [4]. Resistance has developed against almost all classes of antibiotics, especially antibiotics that are frequently used to treat common bacterial infections [5]. Resistance to third-generation cephalosporins and fluoroquinolones was observed in *E. coli* across five WHO regions and in *K. pneumoniae* across six WHO regions [6]. According to the Global Antimicrobial Resistance and Surveillance System (GLASS) report of 2022, it is stated that there is an increase in empiric carbapenem use due to the increasing resistance that has been seen in *K. pneumoniae* to third-generation cephalosporins [7]. In two of the WHO regions, there was a rate of carbapenem resistance of more than 50% [6].

It is very important that antibiotics are only given to patients who really need them and that last-resort antibiotics be preserved [1]. Difficult-to-treat resistant organisms show resistance to most first-line antimicrobials, like fluoroquinolones and β-lactams, including carbapenems, and treatment of these organisms is becoming an enormous challenge [8].

Carbapenems are one of our last-line antimicrobials; therefore, it is important that we use them correctly [9]. Carbapenems are ideal for treating resistant pathogens since they are not easily broken down by most β-lactamases and extended-spectrum β-lactamases [10]. The WHO has placed carbapenems in the WATCH classification, and these antibiotics are known to have a high chance of resistance [11]. In the last few years, the use of carbapenems has increased and this has caused an increase in Gram-negative bacteria that are resistant to carbapenems [12]. In a global point prevalence survey performed in 53 countries, it was found that carbapenems were the most prescribed antibiotics in the 12 African hospitals which took part in this survey [13]. A study that was conducted in the United States of America (USA) found that the unwarranted use of carbapenems in hospitals can increase the rates of carbapenem resistance amongst complicated Gram-negative bacteria like *P. aeruginosa* [14]. Another study that was conducted in Greece found that there is a proven positive relationship between the use of carbapenem and carbapenem resistance [15,16]. In a study that was conducted in Africa, it was found that the contributing factors for infections with carbapenem-resistant enterobacterales (CREs) were the previous use of antimicrobials, indwelling devices, a prolonged hospital stay and intensive care unit admission (the same factors reported worldwide); 54,4% of these studies were performed in North Africa [17].

The Centres for Disease Control and Prevention (CDC) in the USA describes carbapenem-resistant Gram-negative bacteria as “posing an urgent threat to global health” [18,19]. Therefore, the aim of ASPs is to improve carbapenem utilisation, which includes encouraging a de-escalation strategy to reduce carbapenem usage [14].

In 2016, the Heads of State of the United Nations General Assembly sanctioned a political declaration to address the global problem of AMR [20]. The declaration stated that AMR requires a global response since it is threatening the achievement of sustainable development goals [6]. A National AMR Strategy Framework was drafted in South Africa in 2015 by the NDoH, who acknowledged the threat of AMR. This is a comprehensive approach with roles and responsibilities that are aimed at addressing AMR [20]. Minimising the occurrence of AMR is the main goal of ASPs [21]. Studies conducted in Palestine have shown that ASPs have a positive influence on correct antimicrobial prescribing and on the duration of hospital stays [22]. Antimicrobial stewardship programmes can refine the use of antimicrobials by using different strategies [23], one of which is de-escalation [21].

The aim of de-escalation is to decrease the use of broad-spectrum antimicrobials based on the cultures received and the sensitivity [24]. Some of the final anticipated outcomes for de-escalation include adapting the treatment in accordance with the clinical response; adapting the antibiotic treatment according to the results of the microbiological culture and/or susceptibility of the given bacteria; and ceasing treatment and/or changing therapy from more than one antibiotic to monotherapy [25,26]. Targeted therapy refers to antimicrobial treatment that is modified based on microbiological culture and sensitivity results of the patient. It is very important, but is only possible when the causative organism is provided by cultures with an antibiogram, which is a long procedure that can take up to 24 h or more. Therefore, treatment with empiric antibiotics needs to be initiated [27]. There are many factors that can influence the choice of empiric therapy, including the most likely pathogen, resistance patterns (based on local data and antibiograms), the degree of illness, the infection site and the co-morbidities of the host [28]. The level of infection markers can further guide decisions on whether to de-escalate or discontinue treatment. These markers, commonly referred to as biomarkers, include C-reactive protein (CRP), which is affordable and used in intensive care units for patients with infections that are life-threatening [29], and procalcitonin (PCT), which is commonly used to help diagnose sepsis or bacterial infections [30]. Other biomarkers that can be used are white blood cells (WBCs) and the erythrocyte sedimentation rate (ESR). However, one cannot completely rely on biomarkers as they are not exclusively specific to infections. They can be elevated in non-infectious inflammation such as autoimmune conditions, trauma and surgery. As Matuszak highlights, these non-specific elevations increase the risk of misinterpreting underlying symptoms as indicative of infection, which may subsequently contribute to the unnecessary or prolonged use of antimicrobials [31].

Once empiric antibiotic therapy has been started, the usefulness of the chosen empiric treatment should be assessed for the opportunity to practice de-escalation or to discontinue the antibiotic therapy [32]. Even though de-escalation is one of the main elements of antimicrobial stewardship (AMS), it remains the most challenging [21]. It is recommended to switch to an active antimicrobial when the microscopy, culture and sensitivity results show that an antimicrobial is not active against the causative organism and was started as empiric therapy [33]. De Waele et al. refer to antimicrobial de-escalation as the “streamlining of antimicrobial therapy” [34]. This means if there is a lack of a suitable narrow-spectrum alternative such as in cases where the cultured organism shows resistance to multiple antibiotic classes, the carbapenem will still be kept as the streamlined antimicrobial for that patient. If an ESBL or MDR organism is present, carbapenems are the first-line therapy; therefore, de-escalation will also not be conducted. In critically ill patients like those with sepsis and complicated infections, ongoing carbapenem therapy regardless of culture results is necessary due to the high risk of clinical deterioration. Effective de-escalation leads to a decrease in AMR and antibiotic-related side-effects, as well as a decrease in antimicrobial costs [35]. Some of the barriers to de-escalation include, but are not limited to, a shortage of diagnostic facilities, a lack of education and multidisciplinary collaboration and the indecision around de-escalating antimicrobials in patients who are critically ill and who are clinically improving with the use of broad-spectrum antimicrobials [25,36]. If antibiotic de-escalation is not practiced for all eligible patients, there will be an increase in the overuse of antibiotics which will lead to an increase in antimicrobial resistance [36].

Therefore, the objectives of this study were to determine the prevalence of carbapenem prescribing; to assess which carbapenems were prescribed the most; to determine how many patients were de-escalated from carbapenem therapy; and to investigate the reasons why de-escalation was not practiced in others.

## 2. Materials and Methods

Study design, site and period: This study followed a retrospective, descriptive research approach. It was a quantitative study determining the number of patients who were prescribed carbapenems, how many of the prescribed carbapenems were used as empiric treatment, and how many patients on empiric treatment were de-escalated, and if not they were de-escalated, whether a reason could be obtained. This retrospective study was conducted for three consecutive months (16 February 2023–15 May 2023) in a private hospital in South Africa with a bed capacity of 231 patients.

Sample selection and study population: Purposive sampling was used as the researchers deliberately selected participants based on specific characteristics and relevance to the study. The files of all the adult patients who were prescribed carbapenem therapy and were admitted to the hospital during the study period were reviewed. A sample size of 243 patients per month was calculated using Raosoft. The Raosoft Online Sample Size Calculator is a free web-based tool that helps researchers quickly determine the minimum number of survey respondents needed to achieve statistically valid results. By entering values such as population size, margin of error, confidence level, and expected response distribution, the calculator computes the recommended sample size. This ensures that surveys or studies are neither under-sampled (leading to unreliable results) nor over-sampled [37].

Inclusion criteria: All patients aged 18 and above, admitted to the designated wards (intensive care unit (ICU), high care, medical and surgical wards), for which data access was granted, and who were started empirically on carbapenem therapy, were included. The records were accessible for the three-month data window approved by the hospital. The documentation contained the specific clinical parameters outlined for the review. Access was only approved for a three-month period dated from 16 February 2023 to 15 May 2023, and all these patients were reviewed.

Exclusion criteria: All patients that were discharged/deceased within 3 days of starting carbapenem therapy were excluded, as were patients whose records fell outside the three-month data access window granted by the hospital and those not admitted to the specified ward or department for which data access was approved. Documentation did not include the specific parameters outlined in the inclusion criteria. They were diagnosed with conditions other than those permitted for review under the hospital’s data access restrictions. Records were incomplete or lacked sufficient information for analysis; this was to ensure that the final dataset was accurate, reliable and suitable for analysis.

Data collection and data collection instruments: Data was collected by retrospectively reviewing the patient files. The hospital uses software that compiles different reports that contain all medical reports and patient details called SAP, SAP S/4HANA Cloud (SAP integrates verified clinical and administrative records). Data from SAP software was used to identify all eligible patients who received carbapenem treatment during the study period. The files of the patients were then used to complete the data collection sheet. A standardised data collection sheet was adopted from the study of Sadyrbaeva-Dolgova et al. [26], which was used to ensure consistent and reliable extraction of the required variables. The tool was then tested for relevance and clarity by conducting a pilot study on 10 patients who were initiated on carbapenem empiric therapy who were not within the data collection period for the study. The main researcher was the one who collected all the data to ensure reliability and consistency throughout. Each carbapenem “course” was captured on a new data sheet. The data sheet contained the following information: the patient study number, the basic demographical information (age, gender), date of admission and discharge, the reason for admission, antibiotics used in the last three months (if known), infection markers (only PCT and CRP were considered as these are the ones commonly used in the facility of study) on the day carbapenems were started and on day 3 of carbapenem use, if cultures were conducted and the susceptibility thereof, and if the carbapenem treatment was de-escalated or not and to which antimicrobial therapy it was de-escalated to. De-escalation was considered as stopping antimicrobial treatment, de-escalating to an antimicrobial with a narrower spectrum, or changing treatment to targeted therapy (whether it was broader- or narrower-spectrum).

Ethical Approval: Ethical approval for the study was obtained from Sefako Mkgatho Health Sciences University Research Ethics Committee (SMUREC/P/331/2020:PG) as well as from the head office of the participating private hospital. Throughout data extraction, patient confidentiality was protected by using an anonymised data collection sheet and omitting all personal identifiers. Only the authors had access to the data, which were stored securely and used for the purpose of this study.

Statistical analysis: All the data sheets that were collected were captured on a Microsoft Excel^TM^ spreadsheet and double-checked to ensure accuracy. Data was cleaned by reviewing the records to identify and correct any inconsistencies, removal of duplicate entries, and ensuring completeness and accuracy before being sent to a statistician for further analysis. The variables were summarised using a range of statistical measures, including frequencies, percentages, mean, standard deviation, median and interquartile range, depending on the distribution of each variable. Associations between patients whose therapy was not de-escalated and those who were de-escalated and various categorical variables were examined using Pearson’s chi-square test, while independent *t*-tests were employed to compare continuous variables between the de-escalation and non-de-escalation groups. The significance level was set at a *p*-value of 0.05, as the study was conducted with a 95% confidence interval. A *p*-value of <0.001 indicates that if there was a positive organism cultured, de-escalation of therapy was initiated. All analyses were conducted utilising IBM SPSS v28. The data was presented in the form of graphs, charts and tables.

## 3. Results

### 3.1. Population and Prevalence of Carbapenem Prescription

During the study period, approximately 1 754 patients were admitted to the wards included in the study, with a total of 197 (11.2%) patients started empirically on carbapenems, but 21 patients were excluded because they met part of the exclusion criteria.

The study included 176 patients with a median age of 66 (IQR 54–77) years; 53.4% were female and 46.6% were male. The median length of stay in the hospital stay was 13 days for all patients. A total of 77 patients (43.8%) received some type of antimicrobials within the past 90 days. Table 1 shows the characteristics of all patients that were included in the study.

### 3.2. Carbapenems Used for Empiric Therapy

Ertapenem was the carbapenem prescribed the most (62.4%), followed by meropenem (25.4%) and then imipenem/cilastatin (12.2%).

### 3.3. Culture Availability

Cultures were conducted in 125 patients (71.1%), and 68 (38.6%) showed positive organism growth. Some patients cultured more than one organism, giving a total of 75 organisms that were cultured.

*E. coli* was the most cultured organism (37.3%), followed by *K. pneumoniae* (26.7%). From all the positive cultures that were obtained, the Enterobacteriaceae family was cultured in 80%. Table 2 shows whether cultures were obtained or not, as well as the results of the cultures. Knowing if cultures were obtained and whether the patient’s therapy was de-escalated or not will lead us to understanding why patients with positive cultures are not being de-escalated.

### 3.4. De-Escalation

De-escalation was practised in only 30 patients (17%). De-escalation was practiced in 93.3% (*n* = 30) of patients who had culture results (*p* = 0.003) and 92.9% (*p* ≤ 0.001) of the patients who had positive cultures confirmed. Table 3 reports on the de-escalation practises done according to the organisms that were cultured.

De-escalation occurred in 26 patients where growth was seen on the cultures, but there was a remarkable 49 patients who were not de-escalated and for whom no cultures were performed. Figure 1 shows the de-escalation patterns where cultures were performed.

Table 4 shows how empiric therapy was de-escalated (stopped, narrower-spectrum or targeted therapy). Targeted therapy, after cultures were received, was conducted in 21 patients (70%) and some of these patients required a carbapenem (20/21 patients) or a reserve antimicrobial like tigecycline (1/21 patients) since they had an infection by an MDR organism. The antimicrobial that was used most commonly as targeted therapy was meropenem (59%).

### 3.5. Factors or Reasons for Not De-Escalating

The infection markers of the patients who were included in the study were obtained to see if there is a correlation between infection markers and de-escalation. The mean CRP of all patients started on empiric carbapenem treatment was 157.1 mg/dL, and on day three of treatment, the mean CRP decreased to 112.4 mg/dL. Table 5 shows the infection markers of all the patients in the study sample vs. patients who were de-escalated and those who were not.

The most common reason for not de-escalating was that patients were improving clinically on empiric carbapenem therapy, with 67 of the patients (46%) showing a decrease in infection markers, which is viewed as patient improvement. The second most common reason for not de-escalating was the unavailability of cultures in 32% of patients. Figure 2 gives a schematic illustration of the reasons de-escalation was not practiced.

## 4. Discussion

To the best of our knowledge, this is the first study that has been conducted to determine why prescribers do not practice empiric carbapenem therapy de-escalation in South Africa. This is similar to the study by Abuelshayeb et al., who observed that only 40.3% of patients were successfully de-escalated out of the 196 that were eligible for de-escalation [38]. The main finding was that prescribers did not want to de-escalate from empiric therapy, firstly, because according to the biomarkers and presentation, the patient was improving on the current carbapenem therapy and, secondly, because no cultures were performed to guide therapy. When looking at the *p*-values of the study, it can be stated that the availability of cultures (*p* = 0.003), especially positive cultures, is one of the main factors that will lead to antimicrobial de-escalation (*p* ≤ 0.001). If no cultures are available, there is a minimal chance that de-escalation will be practiced. This behaviour is common in settings with high burdens of multidrug-resistant (MDR) Gram-negative infections and limited diagnostic capacity [39]. Antimicrobial stewardship practices need to strengthen diagnostic stewardship by ensuring that culture results are available early enough to guide therapy modification [40]. Education of the physicians is necessary to address the misconception that clinical improvement alone warrants continuation of carbapenems, even when narrower agents may be effective. Without addressing these behavioural and infrastructural factors of diagnostics, efforts to reduce unnecessary carbapenem use to reduce AMR will remain limited.

The right choice of empiric therapy is very important since it can lead to improved treatment outcomes and help to limit AMR [28]. The first objective was to determine the prevalence of empiric carbapenem prescription. This study found that one-tenth of all patients admitted were started empirically on a course of carbapenem treatment. Carbapenem de-escalation is a critical component of antimicrobial stewardship, considering the rise in global antimicrobial resistance [41]. As carbapenems are considered as reserve antibiotics by the WHO, as they help to treat multidrug-resistant infections [42], their overuse can increase the emergence of carbapenem-resistant organisms, limiting future treatment options. De-escalating from carbapenems to narrower-spectrum agents when cultures are obtained not only preserves their efficacy but also reduces the risk of adverse effects and healthcare costs. This encourages targeted therapy and will lead to improved patient outcomes [41].

This is a concern and is commonly seen in private hospitals in South Africa [43]. Despite South Africa’s efforts to reduce antimicrobial resistance (AMR), unnecessary antibiotic prescription rates remain high in both the public and private healthcare sectors [43]. Notably, private providers tend to prescribe antibiotics with a higher risk of resistance development, and our study setting was a private hospital. There are inconsistent practices between sectors, poor adherence to national treatment guidelines, and significant gaps in AMR education among healthcare workers and students [44]. These challenges neglect the country’s stewardship initiatives and highlight the urgent need to standardise protocols and decrease the spread of resistant infections.

Members of the Enterobacteriaceae family were the most prevalent organisms that were cultured, with an occurrence rate of 77.4% in all positive cultures. *E. coli* was the most prevalent pathogen, accounting for 31/84 isolates, followed by *K. pneumoniae*, which accounted for 21/84 isolates. This is similar to the information that was submitted by the NDoH, where it was found that Enterobacteriaceae *K. pneumoniae* was the most prevalent organism cultured from blood results, followed by *E. coli* [3]. This is a concern, as according to the Global Antimicrobial Resistance and Surveillance System (GLASS) report of 2022, the increasing resistance that has been seen in *K. pneumoniae* to third-generation cephalosporins is causing an increase in empiric carbapenem use [7]. Ertapenem was the empiric carbapenem of choice for 62.4% of the patients in our study, similar to other studies performed in the same setting [45]. Other studies have shown that this is usually the carbapenem of choice in patients who are not critically ill, who do not have a risk of being infected by *Pseudomonas* spp. or *Acinetobacter* spp. and who have not been treated with other antimicrobials in the past 90 days [46].

C-reactive protein can help to differentiate between bacterial and viral infections as well as to determine the bacterial infection’s severity [47]. The CRP was >100 (raised) in 55.8% of the patients when empiric carbapenem therapy was started and the CRP after 3 days of carbapenem therapy was <100 in 48.2% of patients, which indicated that patients were improving on carbapenem therapy. PCT, which is a more sensitive marker for bacterial infections [48], was raised in 26.9% of the patients at the start of carbapenem therapy, but PCT was not performed often since it is an expensive test. It was not performed in 72.1% of the study population. After 3 days of carbapenem therapy, PCT testing was only conducted in 16.7% of the population. It has been shown that CRP and PCT can help to de-escalate antimicrobial therapy at an earlier stage [49]. Therefore it is crucial to invest in ways in which these tests can be conducted more often. De-escalation was not practiced in most of the patients (73.6%), mostly due to patients’ infection markers improving (decreasing) while on carbapenem treatment (46%), which was seen as clinical improvement—these patients did not have any positive cultures. Other factors, similar to other studies, also played a role in patients not being de-escalated, for instance, infection marker increases or a small change in infection markers after carbapenems had been started [1]. Masterton also listed these as some of the barriers to de-escalation, where it was found that de-escalation was not practiced due to the indecision to de-escalate antimicrobials in patients who are critically ill and who are clinically improving with the use of broad-spectrum antimicrobials [25,36].

The unavailability of cultures was the second highest contributor to patients not being de-escalated, and was seen in 32% of the patients that were not de-escalated. This is concerning because it results in the empirical use of broad-spectrum antimicrobials without definitive identification of the underlying pathogens. This correlates with the study that was performed by Jacob et al., where microscopy, culture and sensitivity tests were performed for only 34% of the patients. Of these patients, who had a positive culture, proper de-escalation only took place in 36% [50]. For de-escalation to take place, cultures are needed to target the correct organism with the correct antimicrobial [21]. This was a challenge, as cultures were not always performed in this study setting, with 29% of patients not have cultures taken. Of the 71% of patients that had cultures taken, only 54.3% of cultures had bacterial growth. It is common to obtain high culture-negative rates when empiric antibiotic therapy is started before cultures are obtained [51]. Obtaining blood cultures after a patient received antimicrobial therapy could result in a lowered blood-culture yield [52].

After the culture results became available, antimicrobial de-escalation was practiced in 15.7% of all the patients who were started on carbapenem therapy (*n* = 197). This is quite a small number compared to the 73.6% that were not de-escalated. In the de-escalation group, 71% of the patients received targeted escalation therapy since they had an infection caused by an MDR organism, mostly *K. pneumoniae*, and carbapenems were the first line of therapy for these organisms. Patients with ESBLs had to stay on the carbapenem or change to tigecycline. Most of these patients were treated with meropenem.

There is also a rise in infections caused by CREs in South Africa, as seen in a study that reported a higher rate of resistance in Enterobacteriaceae like *K. pneumoniae* than in *E. coli* in public facilities [53].

For the 73.6% of patients not de-escalated, there was a resulting increase in the use of carbapenems for longer periods, with an average of six days. A total of 42% of patients received carbapenem therapy for longer than 6 days. Extending the duration of use with broad-spectrum antibiotics like carbapenems is associated with increased risk of antimicrobial resistance [54], higher healthcare costs and potential adverse effects [55]. Decreased rates of de-escalation contribute directly to longer treatment courses, and this can be a missed opportunity for antimicrobial stewardship interventions aimed at optimising antibiotic use and minimising unnecessary exposure to antimicrobials.

Inpatient settings will continue to be an important target for AMS interventions and improving the utilisation of antimicrobials, and this study has shown possible areas where a clinical pharmacist can play a role, similar to other studies where the pharmacist played a role in improving utilisation of antibiotics [56,57]. Pharmacists are seen as custodians of medicine; they are perfectly situated and should play a principal role within a multidisciplinary team to promote and coordinate the implementation and monitoring of ASPs [21]. According to the findings in this study, the pharmacist can play a role in advocating for cultures to be taken, encouraging de-escalation, and collaborating with a multi-disciplinary team to help guide a patient-specific treatment plan. This can be conducted through ASP-led interventions to reduce carbapenem use, encouraging doctors to review and reassess patients continuously. Targeted stewardship rounds can also be conducted with a focus on de-escalation practices.

We are aware of some limitations in this study. This study was conducted in a private hospital and is not representative of SA as a whole. However, this hospital has similar prescribing patterns to most other private hospitals. In the study conducted by Engler et al., it was noted that compliance to the National Strategic framework was lower [58]. This study could be used as an initial study to try and create guidelines for de-escalation to try to implement and reduce the use of carbapenem antibiotics, as they are in the WATCH category according to the AWaRe classification, meaning they are prone to resistance [59].

In a national point prevalence survey conducted in South Africa, antibiotics in the WATCH category were used mostly in the ICU and it is concerning that this practice is now in the whole hospital setting [60].

This was a retrospective study and, therefore, the patient’s clinical picture was not considered and the reasons for not de-escalating were solely based on the laboratory results. Some patients’ files had been closed and new files opened—this was not considered for the length of hospital stay. The same patient who was started on a new carbapenem course was seen as a new carbapenem prescription; this may increase the number of empiric therapies. One other limitation is the exclusion of patients who died during the therapy, as this introduces sampling bias. Despite these limitations, we believe our findings are robust, providing direction for the future on de-escalation. This includes a key role for clinical pharmacists in progressing antimicrobial stewardship programme activities in critical patients admitted to the hospitals. We would recommend for future studies to be conducted more widely and for more de-escalation to be conducted.

Although this study has limitations, its findings are still highly relevant, because antimicrobial overuse is well documented in South African private hospitals, where broad-spectrum agents are more readily available and prescribed. Therefore, the low de-escalation rate observed may represent a broader, system-wide stewardship challenge rather than an isolated hospital-specific issue.

This highlights the value and importance of the current study as it provides early evidence of suboptimal de-escalation, thereby underscoring the urgent need for strengthened AMS interventions across all the healthcare system.

## 5. Conclusions

The increasing use of carbapenems as empiric therapy is of concern considering the increasing rate of AMR. Prescribers were reluctant to de-escalate treatment once they saw that their patient was clinically improving; however, this contributes more to the increased use of antibiotics, which leads to increased antimicrobial resistance. Considering that carbapenems are also part of the WATCH antibiotics, this is a great concern. We recommend initiating empiric antimicrobial therapy based on patients’ symptoms, medical history, and local epidemiology while awaiting culture results, because without cultures, de-escalation is not feasible and the treatment is “a shot in the dark”. With the growing rate of AMR, de-escalation will become of more importance to preserve the antibiotics that still work. Rising empiric carbapenem use drives antimicrobial resistance, highlighting the urgent need for stewardship interventions to enable timely, safe de-escalation. Future studies to investigate barriers to de-escalation, including knowledge gaps, patients’ fear of clinical deterioration and infrastructure limitations, need to be performed.

## Figures and Tables

**Figure 1 antibiotics-14-01220-f001:**
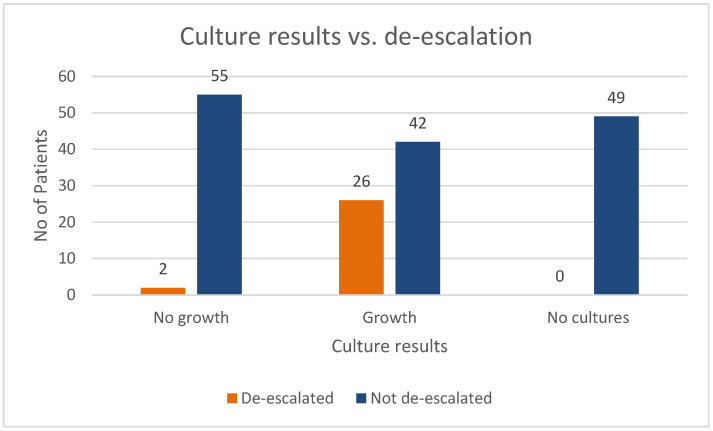
Culture results versus de-escalation patterns.

**Figure 2 antibiotics-14-01220-f002:**
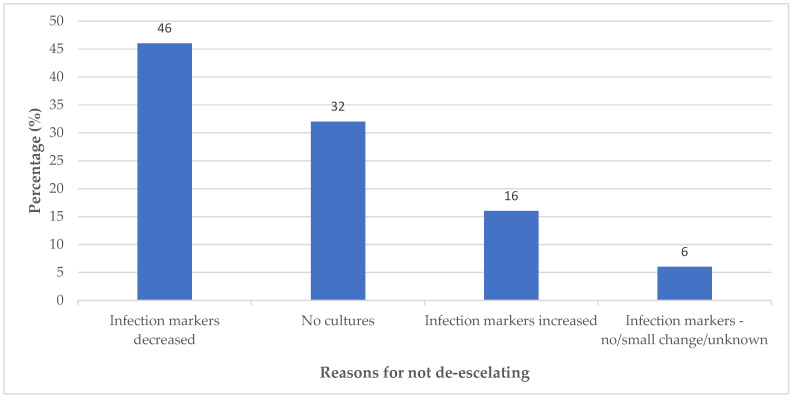
Reasons for not de-escalating.

**Table 1 antibiotics-14-01220-t001:** Characteristics of patients who received empiric carbapenem therapy.

Variables	Total (*n* = 176, %)	De-Escalated (*n* = 30, %)	Not De-Escalated (*n* = 146, %)	*p*-Value
Age (years)				0.466
20–29	5 (2.8)	0(0)	5 (3.4)	
30–39	15 (8.6)	5 (16.7)	10 (6.8)	
40–49	13 (7.4)	1 (3.3)	12 (8.2)	
50–59	33 (18.7)	5 (16.7)	28 (19.2)	
60–69	39 (22.2)	7 (23.3)	32 (21.9)	
70–79	41 (23.3)	9 (30)	32 (21.9)	
>80	30 (17.0)	3 (10.7)	27 (18.5)	
Sex				0.232
Male	82 (46.6)	11 (36.7)	71 (48.6)	
Female	94 (53.4)	19 (63.3)	75 (51.4)	
Any antimicrobial use in past 90 days				0.118
Yes	77 (43.8)	17 (56.7)	60 (41.1)	
Unsure	99 (56.2)	13 (43.3)	86 (58.9)	

**Table 2 antibiotics-14-01220-t002:** Culture availability and organisms cultured.

Variables	Total (*n* = 176, %)
Cultures	
Yes	125 (71.1)
No	51 (28.9)
Culture results	
No growth	57 (32.4)
Growth	68 (38.6)
No cultures	51 (29.0)
Organism cultured *Campylobacter* spp. *Pseudomonas aeruginosa* *Streptococcus group* *Staphylococcus aureus* *Staphylococcus epidermidis* Enterobacteriaceae *Citrobacter* spp. *Escherichia coli* *Enterobacter. cloacae* *Klebsiella pneumoniae* *Morganella morganii* *Proteus mirabilis* *Salmonella* spp. *Serratia* spp.	(*n* = 75, %) 1 (1.3) 5 (6.7) 2 (2.7) 3 (4.0) 1 (1.3) 3 (4.0) 1 (1.3) 28 (37.3) 3 (4.0) 20 (26.7) 2 (2.7) 3 (4.0) 1 (1.3) 2 (2.7)

**Table 3 antibiotics-14-01220-t003:** De-escalation with regard to cultures.

Variables	Total (*n* = 176, %)	De-Escalated (*n* = 30, %)	Not De-Escalated (*n* = 146, %)	*p*-Value
Cultures				0.003
Yes	125 (71.1)	28 (93.3)	97 (66.4)	
No	51 (28.9)	2 (6.7)	49 (33.6)	
Culture results No growth Growth	57 (32.4) 68 (38.6)	*n* = 28 (Cultures) 2 (7.1) 26 (92.9)	*n* = 97 (Cultures) 55 (56.7) 42 (43.3)	<0.001

**Table 4 antibiotics-14-01220-t004:** Empiric therapy and de-escalation.

De-Escalation Strategy	(*n* = 30, %)
Antimicrobials stopped	2 (6.7)
Narrower spectrum given	7 (23.3)
Targeted therapy	21 (70)

**Table 5 antibiotics-14-01220-t005:** Patient infection markers.

Variables	Total *n* = 176 (%)	De-Escalated *n* = 30 (%)	Not De-Escalated *n* = 146 (%)	*p*-Value
CRP on the start of treatment				0.220
0–49	24 (13.7)	6 (20)	18 (12.3)	
50–99	34 (19.3)	2 (6.7)	32 (21.9)	
100–200	52 (29.5)	8 (26.7)	44 (30.1)	
>200	48 (27.3)	9 (30.0)	39 (26.7)	
Not known	20 (10.2)	5 (16.7)	13 (8.9)	
CRP 72 h after starting carbapenem treatment				0.617
0–49	47 (26.7)	7 (23.3)	40 (27.4)	
50–99	42 (23.9)	8 (26.7)	34 (23.3)	
100–200	47 (26.7)	6 (20)	41 (28.1)	
>200	27 (15.3)	5 (16.7)	22 (15.1)	
Not known	13 (7.54)	4 (13.3)	9 (6.2)	
PCT on start of carbapenem treatment				0.460
Increased	48 (27.3)	8 (27.7)	40 (27.4)	
Not increased (normal)	2 (1.1)	1 (3.3)	1 (0.7)	
Not available	126 (71.6)	21 (70)	105 (71.9)	
PCT 72 h after starting carbapenem treatment				0.809
Increased	28 (15.9)	5 (16.7)	23 (15.8)	
Not increased (normal)	2 (1.1)	0 (0.0)	2 (1.4)	
Not available	146 (83.0)	25 (83.3)	121 (82.9)	

## Data Availability

Research data can be obtained on reasonable request from the corresponding authors.

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
