# Peer review of "Determining the Use and Reasons for Non-De-Escalation of Empiric Carbapenem Therapy in a Private Hospital in South Africa"

_antibiotics, 2025, doi:10.3390/antibiotics14121220_

Round 1
Reviewer 1 Report
Comments and Suggestions for Authors
- PLEASE SEE COMMENTS IN ATTACHED FILE . PLEASE REPLY
- please mention method of doing identification and AST of bacteria isolated

Author Response
Dear Reviewers,
We sincerely thank you for taking the time to review our manuscript and for providing valuable feedback. We have carefully considered all of your comments and suggestions, and we have attempted to address each point in detail. We hope that our revisions and clarifications adequately respond to your concerns and meet the expectations for improvement.
We greatly appreciate your constructive input, which has helped us strengthen the quality and clarity of our work.
With gratitude,
Authors

Reviewer 2 Report
Comments and Suggestions for Authors
Line 27-28 in abstract is awkward and warrants rephrasing considering its overall importance in stating the conclusions
Line 47 in the introduction about the prevalence of ESBL is also written awkwardly, and reads as if the authors don’t understand the statistical use of prevalence—rephrase
Line 48-53—where does South Africa fall in the WHO regions and why is this data relevant to the study? Please add context for readers
Perhaps the two paragraphs lines 43-60 can be reworked for readability and also to limit redundancy and improve flow, as well as incorporate the paragraph line 85-94.
Line 66—what is “big chance” of resistance? The literature more commonly uses phraseology such as “high rates,” if that is what is being indicated
The introduction is multiple paragraphs too long, wandering around the topic of AMR and providing lots of citations which elaborate on the AMR/carbapenem overuse problem, but not all of the references add additional relevance to the problem of AMR and carbapenem use in South Africa. Would edit aggressively as it is difficult to follow and I have lost interest by the time I realized the introduction was still going on page 3, let alone that it goes on to page 4. Stewards know much of the baseline data and while it is important to reiterate, this is extremely redundant. Focus on the South Africa context.
Methods:
Sample size of 243 patients to provide power? How is this relevant when purposive sampling is used? There is a disconnect here.
“Infection markers” utilized in introduction and then in results needs to be defined in the methods as different physicians will order different “infection markers” and not all consider CRP or procalcitonin relevant “infection markers”; also the results is the first time I see procalcitonin mentioned at all—also consider it makes up a significant proportion of the discussion, absolutely need to be introduced earlier—and would need more robust citations of its utility rather than just bringing them up in the conclusions
Results:
All tables—(n/%) needs to be in the column labels, not next to each line for the entire table
Line 207-208 does not belong in results but in conclusions/discussion as it is a comment on the results. If the point is to add context to the statistics, it needs to be rewritten as a statement about the comparison statistics, not interpreting it for the reader.
Figure 2 needs to be reformatted in decreasing frequency in the columns (46—32—16-6 not with 6 ahead of 16)
Discussion:
P values should be stated when referencing them (line 247)
Line 257 should be in the introduction and re-written to add context from this study and why the results of the study are relevant to mention it here
Line 308-309 is irrelevant as readers should know the spectrum of activity of mero vs erta, and is not relevant unless authors want to comment on Pseudomonas or Acinetobacter cultures in this study.
Lines 317-320 are introductory material, not conclusions
Line 323 “CRE’s” is grammatically incorrect, no apostrophe needed
Line 328-329—why are you commenting on duration of therapy without adding context to your study?
Exclusion criteria adds sampling bias for those patients who died—this is appropriate but a limitation that is not mentioned in the conclusions
Comments on the Quality of English LanguageMultiple grammatical errors that need to be addressed, as well as readability comments I placed in the review itself
Author Response

(The authors gave the same response as above.)

Reviewer 3 Report
Comments and Suggestions for Authors
Dear Authors,
Thank you for submitting the manuscript titled "Determining the use and reasons for non-de-escalation of empiric carbapenem therapy in a private hospital in South Africa".
Your manuscript is well written but I have the following recomendation for you to icrease the readibilty and the quality of your paper
Overall, the manuscript is well-structured and provides valuable insights into antimicrobial stewardship practices in the South African private healthcare setting.
Typographical and Minor Errors:
Line 24: "treantment" should be corrected to "treatment."
Line 50: Sentence structure around "Resistance to third-generation cephalosporins and fluoroquinolones was seen in Escherichia coli, in five of the WHO regions and to Klebsiella pneumonia, in six WHO regions ." Consider rephrasing for clarity. Suggest: "Resistance to third-generation cephalosporins and fluoroquinolones was observed in Escherichia coli across five WHO regions and in Klebsiella pneumoniae across six WHO regions."
Lines 40-41: On "Escherichia coli, followed by Staphylococcus aureus, Klebsiella pneumoniae..." consider replacing commas with semicolons or restructuring for better readability because of the list length.
Line 145: The phrase "A report was drawn from SAP to identify all the patients who received carbapenem treatment in our study period." could be more precise: "Data from SAP software was used to identify all patients who received carbapenem treatment during the study period."
Inconsistency in naming organisms: "K. pneumonia" should consistently be "Klebsiella pneumoniae."
Line 201, footnote “a – Only two patients were included in this category” appears detached from context and should be clearly linked to the relevant table or content.
Line 223: "mg/dL" should be consistently formatted throughout; sometimes it appears with a space (mg/dL), sometimes without.
Overall: Some sentences are lengthy and could be split for easier readability.
And I have also more recommendations as below;
1. Add more recent literature references explicitly related to carbapenem stewardship and de-escalation initiatives in similar settings or LMICs.Enhance the explanation of AMR epidemiology specific to South Africa, emphasizing private vs public sector antimicrobial use differences.
2. Clarify the inclusion and exclusion criteria—especially clarify why patients discharged or deceased within three days were excluded and how this may impact data interpretation. Provide more detail on the "data collection sheet" adaptation and how data validity was ensured. Consider adding detail about ethical clearances or patient confidentiality handling earlier (currently appearing in discussion).
3. Include clearer explanations for all p-values reported, specifying which tests were used (e.g., Pearson’s chi-square, t-test) for each association.Provide some additional visual summaries such as bar charts or pie charts for key findings (e.g., culture positivity proportion, reasons for non-de-escalation). Clarify “targeted therapy” subgroup and why most patients remained on carbapenems despite de-escalation.
4. Expand on implications of findings for antimicrobial stewardship programs (ASPs) in South Africa and similar healthcare contexts. Highlight potential interventions the hospital or health system can adopt to increase culture collection rates and support de-escalation. Discuss limitations with greater emphasis on how they influence generalizability to other hospitals and the public sector.
5 Suggest future prospective or interventional studies.
6. Improve sentence flow by shortening lengthy sentences. Ensure consistent terminology for bacterial species, infection markers (CRP, PCT), and antimicrobial agents. Proofread for minor grammar issues and typographical mistakes *I noted above.
I Believe with these revisions, the manuscript will significantly improve in clarity and impact.
Best Regards
Author Response

(The authors gave the same response as above.)

Reviewer 4 Report
Comments and Suggestions for Authors
This study investigated why prescribers rarely de-escalate from empiric carbapenem therapy, a key contributor to antimicrobial resistance. In a retrospective review at a South African hospital, de-escalation occurred in only 17% of cases, mainly due to clinical improvement on therapy or lack of culture results. The study highlights critical barriers to de-escalation, offering insights to improve antimicrobial stewardship practices. However, the study’s relevance appears mostly local, as it focuses on patients from a single hospital. It also acks clearly defined research objectives. Its retrospective design meant that patients’ clinical conditions were not considered, and the reasons for not de-escalating were based solely on laboratory data. Furthermore, the discussion does not sufficiently compare the findings with similar studies from other countries, limiting the study’s broader contribution to understanding de-escalation practices.
Author Response

(The authors gave the same response as above.)

Round 2
Reviewer 2 Report
Comments and Suggestions for Authors
Overall improved with edits.
Author Response
Thank you so much for all the comments, the manuscript has trully been improved.
Reviewer 3 Report
Comments and Suggestions for Authors
Dear Authors
Thank you for submitting revised version.
Best Regards
Author Response
Thank you so so much for all the comments. They were appreciated and the manuscript has been improved.
Reviewer 4 Report
Comments and Suggestions for Authors
Dear Authors,
Thank you for your detailed and comprehensive responses to the reviewer comments, as well as for the substantial improvements made to the manuscript. Your revisions have strengthened the study considerably, particularly in terms of methodological clarity, interpretation of findings, and overall structure.
While the manuscript is much improved, a few remaining details still need attention.
Line 122-123: Please rewrite the sentence explaining the surname Matuszak.
Line 162: Please rewrite the sentence and explain Raosoft online sample size calculator and cite it.
Table 1 and Table 5: Please adjust the layout so that Tables appear entirely on one page for better readability and easier data interpretation.
Line 349: Please remove the gap between two sentences and a between the full stop and the bracket of the previous sentence.
Line 431 and Line 460: Please make the use of the WATCH acronym uniform throughout the manuscript
Given the critical importance of antimicrobial resistance (AMR) in many African healthcare settings, your work addresses a highly relevant and underrepresented area. Understanding the drivers of non-de-escalation in empiric carbapenem therapy is essential for informing and optimizing antimicrobial stewardship (AMS) strategies in regions where AMR presents a growing challenge.
With the revisions now incorporated, I am pleased to recommend your manuscript for acceptance after minor revision. Thank you for your valuable contribution to the field and for your efforts in improving the paper.
Author Response
Thank you so much and all these comments have made the manuscript better
